# Genetic Diversity and Synergistic Modulation of Salinity Tolerance Genes in *Aegilops tauschii* Coss

**DOI:** 10.3390/plants10071393

**Published:** 2021-07-07

**Authors:** Adeel Abbas, Haiyan Yu, Hailan Cui, Xiangju Li

**Affiliations:** Key Laboratory of Weed Biology and Management, Institute of Plant Protection, Chinese Academy of Agricultural Sciences, Beijing 100193, China; adeel.abbas92@yahoo.com (A.A.); yuhaiyan2103@163.com (H.Y.); hlcui@ippcaas.cn (H.C.)

**Keywords:** *Aegilops tauschii* Coss, stress tolerance, salinity genes, abiotic stress, microsatellite markers

## Abstract

*Aegilops tauschii* Coss. (2n = 2x = 14, DD) is a problematic weed and a rich source of genetic material for wheat crop improvement programs. We used physiological traits (plant height, dry weight biomass, Na^+^ and K^+^ concentration) and 14 microsatellite markers to evaluate the genetic diversity and salinity tolerance in 40 *Ae. tauschii* populations. The molecular marker allied with salinity stress showed polymorphisms, and a cluster analysis divided the populations into different groups, which indicated diversity among populations. Results showed that the expression level of *AeHKT1;4* and *AeNHX1* were significantly induced during salinity stress treatments (50 and 200 mM), while *AeHKT1;4* showed relative expression in roots, and *AeNHX1* was expressed in leaves under the control conditions. Compared with the control conditions, the expression level of *AeHKT1;4* significantly increased 1.7-fold under 50 mM salinity stress and 4.7-fold under 200 mM salinity stress in the roots of *Ae. tauschii*. *AeNHX1* showed a relative expression level of 1.6-fold under 50 mM salinity stress and 4.6-fold under 200 mM salinity stress compared with the control conditions. The results provide strong evidence that, under salinity stress conditions, *AeHKT1;4* and *AeNHX1* synergistically regulate the Na^+^ homeostasis through regulating Na^+^ transport in *Ae. tauschii*. *AeNHX1* sequestrated the Na^+^ into vacuoles, which control the regulation of Na^+^ transport from roots to leaves under salinity stress conditions in *Ae. tauschii*.

## 1. Introduction

Soil salinity is one of the major environmental factors which limits the agriculture productivity [1]. It is estimated that more than 6% of the world’s land area has been affected by salinity [2]. The main consequences under salinity stress conditions are an interruption in Na^+^ homeostasis, abnormality in cellular metabolism, membrane dysfunction, flaws in plant development, impacted photosynthesis, and consequently, slowed plant growth [3,4]. Plants have developed various mechanisms against salinity stress including sequestering Na^+^ concentration in vacuoles, decreasing Na^+^ accumulation in the cytosol, and extruding cytoplasmic Na^+^ out of the cell aided by (bifunctional K:H/Na: H antiporter) *NHX1* [5]. Exploring stress mechanisms focuses on stress perception and stress-induced biochemical, physiological, and genetic changes in plants [6,7].

The *Ae. tauschii* D genome was impaired with Na^+^ exclusion in bread wheat and increases salinity discriminations. The indicative restricting unidirectional Na^+^ movement indicates the salt tolerance mechanism in crops [8]. As a result, Na^+^ is a fundamental strategy for plants to overcome salinity stress, and the related phenotypes, facilitated by different Na^+^ transport-related genes, including *HKT1*, play an important role in vital functions in Na^+^ homeostasis [9]. *HKT* transporter also interferes with Na^+^ transport and is involved in Na^+^ and K^+^ homeostasis [10]. A higher ratio of K^+^/Na^+^ is maintained by a specific Na^+^ transporter (*HKT1;5*) to confer salinity tolerance [3]. In *Arabidopsis thaliana*, the *HKT;1* is responsible for Na^+^ movement in xylem vessels and decreases Na^+^ concentration in leaves [11]. A high expression level of *HKT1;4* in roots boosts the inflow of Na^+^ in XPCs, which restrains Na^+^ accumulation in leaves [12]. Proteins located on plasma membranes of surrounding cells remove Na^+^ from the xylem, thus reducing its transport towards plant leaves [13].

In the regulation of Na^+^ homeostasis, *AeNHX1* plays an essential role by sequestering Na^+^ into plant vacuoles [14]. It is well documented that overexpression of the *AeNHX1* gene plays a vital roles in salinity tolerance of different plant species, such as *Arabidopsis*
*thaliana* and *Oryza sativa* [15]. Similarly, the expression of *NHX1* is upregulated in *A. thaliana* under salinity stress conditions [16]. *HKT1;4* is active in *Zygophyllum xanthoxylum* under mild salinity stress. This *HKT1;4* leads to the deposition of Na^+^ into the xylem, which then transmits Na^+^ to the leaves, and lastly plays a key role in Na^+^ compartmentalization in the plant leaves [17]. *HKT1;4* shows Na^+^ exclusion from the blades during salt stress. Silencing *AeNHX1* changed the typical salt-accumulation characteristics to salt-exclusion, namely, *HKT1;4* became more active and unloaded Na^+^ from the xylem into XPCs, limiting salt accretion in shoots [18]. *HKT1;4* transporters confirmed variability in Na^+^ transport within different cereals [19] (overall, *HKT1;4* and *NHX1* maintain salt-accumulation features in different plants).

*Ae. tauschii* is one of the wheat relatives and is an important source of abiotic stress tolerance genes [20]. *Ae. tauschii* is distributed in the Mediterranean region, present in Syria, Iran, Russia, Kazakhstan, Afghanistan, Pakistan, Turkey, and Iran, and extends eastwards of the Yili Valley of Xinjiang in China [21]. In China, *Ae. tauschii* has been reported in more than ten provinces, and its growth affects wheat crops on a large scale [22]. Genetic diversity between different *Ae. tauschii* populations have been verified using various techniques, including simple sequence repeat (SSR) [23]. Mostly, researchers have reported that salt tolerance varies in different crop species and at different developmental stages [24]. The D genome of *Ae. tauschii* is a rich source of genetic material, and it may serve as a rich source of genes, such as those involved in salt tolerance, for wheat crop varieties [25]. No adequate work has been published to reveal the genetic diversity of *Ae. tauschii* populations of China through molecular markers allied with salinity stress treatments. Thus, we evaluated the salinity tolerance populations based on physiological parameters and microsatellite markers. We also analyzed the expression profile of *AeHKT1;4* and *AeNHX1* in *Ae. tauschii* under different salinity treatments. Then, we created a model to understand the mechanism of salinity tolerance in *Ae. tauschii.*

## 2. Results

### 2.1. Physiological Traits

Analysis of variance of dry weight biomass, plant height, Na^+^, K^+^, and K^+^/Na^+^ concentration showed significant discrimination when *Ae. tauschii* populations were exposed to salinity stress conditions compared with control. Salinity stress significantly affected all traits except plant height (cm), while a combination of salinity stress with populations showed significant results in all traits (Table 1). Salinity stress affects all the physiological traits of *Ae. tauschii*. Moreover, out of 40 populations, ten populations showed lower Na^+^ concentrations than the rest. Maximum Na^+^ concentration was recorded in population 12 from Shanxi Province, while the minimum (Figure 1) was recorded 39.3 in population 7 collected from Shandong; under salinity stress conditions, 11 populations from different provinces maintained lower concentrations of Na^+^ and high dry biomass (Figure 2). These 11 populations maintained higher survival rates under salinity stress treatments (200 mM NaCl). Variations among populations were observed in Na^+^ and K^+^ accumulation under salinity stress conditions (Figure 1). Na^+^ concentration showed variation in *Ae. tauschii* populations under salinity stress (200 mM NaCl) conditions. Some populations (P) (P.1, P.2, P.4, P.6, P.7, P.8, P.11, P.15, P.17, P.18, P.19, P.23, and P.24) maintain lower concentrations as compared with others. Na^+^ concentration was recorded 6-fold lower in population 12 (from Shanxi Province) than the mean value of other populations under salinity stress treatments (Figure 1). The plant height of individuals in all populations decreased under salinity stress conditions compared with the plants grown under control conditions.

The data recorded about the minimum and maximum plant height showed variation under salinity stress conditions (Table 2). Some of the populations that survived better under salinity stress conditions had increased plant height compared with the plants grown under control conditions. In addition, a relationship was observed between physiological traits and salinity tolerance; under the salinity stress treatment, dry weight biomass, plant height, Na^+^, K^+^, and K^+^/Na^+^ concentrations were correlated with salinity tolerance in *Ae. tauschii* (Table 3).

The linear regression analysis between Na^+^ concentrations and dry biomass showed that populations (P.) (P.1, P.2, P.3, P.4, P.6, P.8, P.9, P.11, P.15, P.17, P.18, P.19, P.23, and P.24) with higher biomass maintained the lower concentration of Na^+^. These populations possessed lower Na^+^ content under salinity stress treatments as compared with other populations included in this study. The regression coefficient between the Na^+^ concentration and dry biomass was related to salinity tolerance (R^2^ = 0.62) under salinity stress treatments (Figure 2).

### 2.2. Molecular Markers

Out of 26 EST-SSR primers used, 14 primers created a clear pattern and showed polymorphism in 40 populations. A total of 60 alleles were amplified using the 14 primers, with an average of 4.28 alleles per primer. (Table 4). Eight alleles were amplified in *Xgwm 410*, *Xgwm 312*, and *Xgwm 3* primers. The maximum polymorphism information content (PIC) was recorded at 0.92 in primer *Xgwm 410*, with an average of 0.39. The highest major allele frequency was 0.98 in the *Xgwm 609* primer, with an average of 0.68. On the basis of phylogenetic analysis (Figure 3), 40 populations of *Ae. tauschii* were divided into four groups; populations with similar genetic relationships were present in the same group. Group I had six, group II had fourteen, and group III and IV had ten populations in each group. Furthermore, the populations that showed superior tolerance under salinity stress conditions were present in the same cluster and showed a similar genetic relationship. Populations (P.1, P.2, P.7, P.8, P.9, P.15, P.16, P.18, P.19, P.23, P.24) that showed salinity tolerance based on physiological response were present in the same group, which showed a close relationship between these populations. The circle indicated that salinity tolerance populations based on physiological parameters were present in the same group, which showed a close relationship between these populations.

### 2.3. HKT1;4 and NHX1 Expression in Ae. tauschii Shoots and Roots

We determined the tissue-specific expression levels of *AeHKT1;4* and *AeNHX1* in *Ae. tauschii* roots and leaves under control conditions. We performed real-time PCR to analyze the relative expression pattern of these two genes. The expression level of *AeHKT1;4* showed high values in roots compared with leaves (Figure 4A,B). The relative expression level of *AeNHX1* was higher in leaves as compared with roots.

### 2.4. Expression Patterns of HKT1;4 in Roots

The expression pattern *of AeHKT1;4* was investigated in roots under different salinity stress treatments (Figure 5) in 10 *Ae. tauschii* populations. Under 50 and 200 mM NaCl conditions, the expression pattern of *AeHKT1;4* was significantly induced compared to the control. The expression level of *AeHKT1;4* was induced under 50 mM NaCl, and 1.2, 1.7, 1.1, 1.4, 1.8, 1.2, 1.4, 1.2, 1.4, and 1.8-fold changes were recorded compared with control conditions. The expression pattern of *AeHKT1;4* was significantly induced under 200 mM NaCl, and 5.6, 5.9, 4.8, 5.8, 4.6, 5.1, 6.1, 4.5, 4.2, and 4.3-fold changes were recorded compared with control conditions.

### 2.5. Expression Patterns of NHX1 in Leaves

The expression level of *AeNHX1* in leaves was investigated under different salinity stress conditions (Figure 6). Under 50 and 200 mM NaCl conditions, the expression pattern of *NHX1* was significantly induced, and 11 populations were used in this experiment. The expression profile of *AeNHX1* showed substantial changes under 50 and 200 Mm NaCl. Conspicuously, in the salinity stress treatment of 200 mM NaCl, the expression level of *NHX1* was higher than the expression level of *AeNHX1* in the 50 mM NaCl treatment. The expression level of *AeNHX1* was 1.5, 1.2, 1.2, 1.2, 1.3, 1.4, 1.1, 1.4, 1.2, and 1.4-fold higher under 50 mM compared with control conditions. The expression level of *NHX1* under 200 mM significantly induced 5.9, 4.3, 5.3, 4.8, 5.4, 5.8, 4.8, 4.9, 5.9, and 4.2-fold higher compared with control conditions.

## 3. Discussion

### 3.1. Perspectives about Salinity Tolerance and Genetic Diversity in Ae.tauschii

In this study, we investigated the physiological parameters and microsatellite markers (allied with salinity tolerance) were investigated under control and salinity stress conditions in *Ae. tauschii*. Our results indicated that all physiological parameters were significantly affected by salinity stress conditions. The most salt-tolerant populations were identified on the basis of dry weight biomass and Na^+^ concentration of leaves under salinity stress conditions. Biomass production in crop plants is one of the most critical and an important factors used to categorize salinity tolerant populations [26]. Eleven populations showed more salinity tolerance compared to the other populations under salinity stress conditions. This may be due to the combined effect of salinity stress and waterlogging conditions in some areas, which may increase in some populations that survived better under hypersaline conditions [27]. The authors [28] have studied *Aegilops cylindrica* populations collected from different countries under salinity stress (400 mM NaCl) treatments. These data illustrate the effects of natural selection on the adaptation of these populations under salinity stress conditions. The gene pool of these ten populations is highly enriched for wheat crop improvement.

Our study’s salinity tolerant populations showed lower dry weight biomass and higher concentrations of Na^+^ under salinity stress conditions among 40 populations. This could be attributed to natural selection, which may have led to the adaptation of these populations to salinity stress conditions. These populations were collected from saline habitats, which support the salinity tolerance of these populations. This phenomenon also explains why most populations that can tolerate salinity have redeveloped mechanisms to thrive in a saline environment. The mechanisms by which plants tolerate salinity stress are the following: (1) osmotic adjustment, (2) attenuation of salt concentration in the plant body through excretion (salt glands) and exclusion (from roots to leaves), and (3) preventing harmful effects on the plant body via compartmentalization [29,30]. A crop plant that maintains low Na^+^ concentration and high dry weight biomass has more salinity tolerance than other populations [31,32]. The present study also reported salt excretion in *Ae. tauschii* populations. Wild relatives of wheat perform better under salinity stress; thus, they can be considered halophytes [33,34].

Microsatellite markers were analyzed to detect genetic diversity among 40 populations of *Ae. tauschii*. The PIC values range from 0.05 to 0.92, showing variation among these populations. Sixty alleles were amplified in fourteen primers, with an average of 4.28 alleles identified per primer. The results obtained in this study were confirmed by a previous study that reported an average of 9.21 alleles per primer and ranged 6–15 alleles per primer that were achieved by SSR markers [35]. The PIC value shows variation because it depends on GT content, the number of alleles per locus, and the type of motifs. The results of the cluster analysis divided the *Ae. tauschii* populations into different groups, which showed genetic variation among these populations. During cluster analysis, populations that showed salinity tolerance were present in one group.

Additionally, the microsatellite salinity tolerance markers *Xgwm 312* and *Xgwm 410* showed polymorphism in *Ae. tauschii* populations. The microsatellite marker *Xgwm* 410 has been previously reported to be linked with salinity tolerance (sodium excretion gene (*NAX2*) from xylem to root). The microsatellite markers *Xgwm 312* and *Xgwm 410* are linked with salinity tolerance and associated with the salinity tolerance gene *HKT1;5* and *HKT1;4* in wheat crops [36,37]. Furthermore, *HKT1;5* and *HKT1;4* are very important for salinity in crops. It was concluded that populations surviving better under high salinity conditions have variability from the normal population. Variability in *Ae. tauschii* populations under salinity stress conditions are vital and helpful for plant breeding.

### 3.2. Role and Expression Pattern of Salinity Tolerance Genes in Ae. tauschii

The gene conferring salinity tolerance mapped to the distal end of chromosome 5AL in wheat corresponds to a Na^+^ transporter with *HKT1;4*, while *Ae. tauschii* is the ancestor of the wheat crop and shares the DD genome [38]. In addition, it supports the availability of a salinity tolerance gene (Na^+^ exclusion). Another marker in our study (*Xgwm 312*) allied with salinity tolerance and closely linked with the gene (*HKT1;4*). *HKT1;4* decreased the rate of Na^+^ movement from roots to shoots and reduced Na^+^ quantity in leaf cells [39]. Furthermore, it showed that the DD genome of *Ae. tauschii* acquired a locus related to the *HKT1;4*. *HKT* plays an important role in regulating Na^+^ and K^+^ transport and maintaining their homeostasis in plants [40]. *HKT* family genes were recognized as Na^+^ transporters in rice and wheat plants and arbitrate Na^+^ reclamation from xylem *HKT1;5* is also expressed in parenchyma cells close to xylem cells to prevent Na^+^ overaccumulation in shoots by blocking Na^+^ transport to leaves. [41]. Furthermore, *AeHKT1;1* unloaded the Na^+^ from the xylem vessels that decreased the Na^+^ concentration in plant leaves, and showed high root expression compared to leaves. *HKT1;4* is regulated by salinity and showed high expression levels in roots but not in leaves of *Triticum monococcum* and *Triticum aestivum* [42]

HKT family proteins in wheat and rice, encoded by *HKT1;4* and *HKT1;5* are acknowledged as Na^+^ transporters and facilitate Na^+^ reclamation from the xylem in wheat and rice. The voltage-clamp investigation showed that to avoid the Na^+^ accumulation in leaves, *HKT1;5* removes excessive Na^+^ from the xylem sap of roots. Likewise, *AeHKT1;1* played a significant role in preventing Na^+^ toxicity by removing Na^+^ directly from xylem vessels, consequently decreasing Na^+^ content in leaves and roots [43]. Similar results have been found for *NAX_2_* and *KNA1* to prevent Na^+^ movement from xylem to leaves. Previous studies reported that *NHXs* play an important role in Na^+^ sequestering and decrease the concentration of cytoplasmic vacuoles in many plants [44]. Overexpression of *ZxNHX* significantly increased the salt tolerance in *Lotus corniculatus* by increasing Na^+^ concentration [45]. *NAX_2_* and *KNA1* were found to be expressed in the roots of *T. monococcum* and *T. aestivum*, respectively, but not in the leaves, and expression levels were upregulated under salinity stress treatments [13]. 

Similarly, *HKT1;4* expressed in the roots of *Ae. tauschii* showed high expression under salinity stress conditions (200 mM), implying that *HKT1;5* played an imperative role in Na^+^ unloading from xylem cells of roots under salinity stress conditions. The authors [46] reported that when barley was sown under salinity stress conditions, the maximum Na^+^ concentration was reached in plant vacuoles, which synchronized *HKT* to remove Na^+^ from the xylem. Our results also showed that Na^+^ accumulation significantly increased in *Ae. tauschii* leaves under salinity stress conditions; these stress conditions strongly induced the expression of *HKT1;4,* thus facilitating the removal of excess Na^+^ from the roots and subsequently alleviating the Na^+^ toxicity in plants. Under salinity stress conditions, one of the fundamental strategies in plants is to reduce the Na^+^ concentration in the cytoplasm through Na^+^ sequestering into vacuoles [47,48].

### 3.3. Synergistic Model of Salinity Tolerance in Ae. tauschii

NHX1 is an omnipresent transmembrane protein that plays an imperative function in compartmentalizing Na^+^ into vacuoles to sustain the Na^+^ homeostasis, increasing salinity salt tolerance in plants [49]. The transcript level of *NHX1* in leaves was noticeably increased under salinity stress conditions in many studies. The transcript level of *McNHX1* increased in the leaves of *Mesembryanthemum crystallinum,* cotton, and chrysanthemum under salinity stress conditions but not in the roots [16,50]. A similar trend was observed in *Z. xanthoxylum* and *D. morifolium* leaves, and a previous study showed a positive correlation between *NHX1* and Na^+^ accumulation under salinity stress conditions [51]. In our results, *AeNHX1* in leaves was pointedly regulated after the 50 and 200 mM NaCl treatments and showed 1.7- and 4.8-folds changes, respectively, compared with the control conditions (Figure 7. Under lower salinity stress conditions (50 mM NaCl), *AeNHX1* in leaves compartmentalized of Na^+^ in vacuoles and sequestering Na^+^ would increase loading into xylem by *AeHKT1;4.* Thus; Na^+^ could transport in leaves by the transportation stream; under high salinity stress conditions (200 mM NaCl), Na^+^ rapidly and unremittingly sequesters in vacuoles of leaves by *AeNHX1*. The vacuole capacity becomes saturated by sequestering Na^+^, which restricted the Na^+^ transport from roots and induced the expression level of *AeHKT1:4* and assists in unloading excessive Na^+^ from the xylem. It also predicted that under the lower concentration of salinity stress, Na^+^ accumulates in plant leaves, and perhaps its concentration is sequestered until vacuole capacity is reached [52]. While under high salinity stress conditions, Na^+^ accumulates in *Ae. tauschii* leaves, which may be why *NHX1* showed high expression compared with control conditions.

Many genes conferring tolerance to salinity stress conditions have been identified in plants. Under salinity stress conditions, plants have complex genetic regulatory mechanisms related to the control of Na^+^ transport and extrusion of Na^+^ from the vacuoles [53]. *HKT1;4*, and *NHX1* were found to be involved in Na^+^ transport and played an important role in salinity stress tolerance. However, *HKT1;4* and *NHX1* have the opposite role in Na^+^ regulation in roots and leaves. In plants, Na^+^ influx and efflux across the plasma membrane of xylem cells contribute to Na^+^ homeostasis [17]. Similarly, in *Z. xanthoxylum*, *NHXI* regulated the Na^+^ accumulation in vacuoles and the Na^+^ transport in the plasma membrane of xylem vessels [18]. Furthermore, *Ae. tauschii* showed lower Na^+^ uptake under salinity stress conditions. In the 50 mM NaCl treatment, *Ae. tauschii NHX1* was induced (Figure 6), following which, Na^+^ was slowly compartmentalized into vacuoles. The high expression level of *NHX1* in leaves showed a large concentration of Na^+^ sequestered in plant vacuoles. *AeHKT1;4* showed a high expression level in roots under salinity stress treatments compared with the control conditions. Under salinity stress conditions, Na^+^ accumulation in leaves was lower because of the sequestration of Na^+^ by *NHX1. HKT1;4* also compensates and becomes involved in Na^+^ loading into the xylem [17].

## 4. Materials and Methods

Forty populations of *Ae. tauschii* collected from five different provinces of China were used in this experiment. Seeds were sown in plastic pots containing gravel under greenhouse conditions. The excess water passed through the hole at the bottom of each pot and collected on the plate underneath. Initially, tap water was applied, and after one week at the second leaf stage, irrigated water was replaced with Hoagland nutrient solution. The Hoagland nutrient solution was added in five gradual steps until the final salt concentrations reached 300 mM NaCl. The study was designed as a randomized complete block with a split arrangement design under salinity treatments (0 and 300 mM NaCl), and 40 populations (Appendix A) from different parts of China were used as a subplot. Three plants were maintained in each pot. After three weeks of treatments, data regarding plant height (cm) and dry weight biomass (g) were recorded. Na^+^ and K^+^ concentrations in leaves were measured by flame photometer (Jenway PFP7, Stone, Staffordshire, UK).

### 4.1. Molecular Markers

For SSR analysis, plant leaves were harvested and stored at −80 °C for DNA isolation. DNA was isolated from leaves using a plant kit (Tiangen Biotech (Beijing) Co., Ltd., Beijing, China). The DNA concentration and quality were checked using 1% agarose gel electrophoresis and a NanoDrop 2000 (Thermo Scientific, Waltham, MA, USA), respectively. Moreover, the DNA concentration was diluted to 30–50 ng μL^−1^ using TE buffer. Out of 26 primers pairs, 14 were used (salinity tolerance SSR primers) for SSR. Polymerase chain reactions (PCRs) were carried out in a 20 μL reaction mixture containing 10 μM PCR MasterMix (0.1 U *Taq* polymerase μL^−1^, 5.0 × 10^−4^ mol L^−1^ dNTPs, 2.0 × 10^−2^ mol L^−1^ Tris-HCl (pH 8.3), 0.1 mol L^−1^ KCl, 3.0 × 10^−3^ mol L^−1^ MgCl; Tiangen, Beijing, China), 40 ng of genomic DNA (1 μL), and 0.6 μM of each forward and reverse primer (7.8 μM). The PCR program consisted of denaturation at 95 °C for 3 min, followed by 35 cycles of 95 °C for 30 s, 45−60 °C (primer temperature) for 30 s, and 72 °C for 2 min, with a final extension at 72 °C for 10 min. Amplified PCR products were separated on 0.9% electrophoreses gels, and the vertical device was used for simple sequence repeats.

The SSR primers with high polymorphism and specific amplification were selected for further study. Among these SSR markers, 14 primers pair were screened and labeled with the fluorescent dyes 6-FAM or HEX by Invitrogen Biotechnology Co., Ltd. (Shanghai, China) (Table 5). Each primer’s annealing temperature was optimized accordingly, and PCR products were verified using 1.5% agarose gel electrophoreses. Primers that displayed polymorphism were screened and labeled with fluorescent dyes; the HUM-STR method was applied for electrophoreses analysis (capillary temperature 60 °C; sample injection 2 KV for 30 s; electrophoresis 4.8 kV, run times: 65 min).

### 4.2. RNA Extraction and cDNA Synthesis

Total RNA was extracted using RNA Plant Kit (Tiangen Biotech Beijing Co., Ltd., Beijing, China). The purity and concentration of RNA were detected using a NanoDrop^TM^ spectrophotometer (NanoDrop Technologies, Wilmington, DE, USA). cDNA synthesis was performed according to the instructions of TransScript Green miRNA Two-Step qRT-PCR SuperMix (Transgenic, Beijing, China). General cDNA was synthesized using the Fast Quant RT kit (Tiangen Biotech Beijing Co., Ltd., Beijing, China) with 1 μg of total RNA and a total volume of 20 μL. The samples were stored at −80 °C until used.

### 4.3. qPCR Analysis

Primer pairs for *AeACTIN*, *AeHKT1;4* and *AeNHX1* were designed using the software Beacon designer with a maximum amplification length of 129 bp (Table 6), an optimal temperature of 55–60 °C, a primer length of 18–22 bp, and a GC percentage of 40–60%. The expression level of *AeHKT1;4* and *AeNHX1* were analyzed in leaves and roots of *Ae. tauschii* under different salinity treatments (50 and 200 mM NaCl). We used 11 populations that perform better under salinity stress conditions. An ABI 7500 qPCR machine was used to detect the gene expression using SYBR green (Applied Biosystems, California, USA). The reaction was conducted in a total volume of 20 μL PCR mix, containing 10 μL Power SYBR Green PCR Master Mix, 1 μL of cDNA, 0.6 μL of each primer, 0.6 μL dye, and 7.8 μL of ddH_2_O. The cycling conditions for the qRT-PCR were set to the following: 10 min at 95 °C, 40 cycles of 95 °C for 15 s and 57–58 °C for 32 s; to obtain a melting curve, the temperature was increased by 0.5 °C every 5 s. The qPCR assays were performed with three technical and biological replicates.

### 4.4. Data Analysis

Analysis of variance (ANOVA) was carried out to examine the effects of NaCl treatments (0 and 300 mM). The statistical analysis of physiological traits and regression analysis between salinity tolerance and Na^+^ concentrations in 40 *Ae. tauschii* populations were conducted by SAS version 9.3 (SAS Institute 2011). The amplified production using fluorescent SSR primers was detected using an ABI PRISM 3730xl DNA sequencer with GS500 (Applied Biosystems, USA) as an internal size standard. GeneMarker version 2.2.0 (Applied Biosystems) was used to determine the allele size. Power marker 3.1 was used to calculate allele frequency, gene diversity, and polymorphism information content. The unweighted pair group method with an arithmetic average (UPGMA) was used to determine the genetic relationships among populations using genetic similarity coefficient values. A cluster analysis was conducted using software (Power marker MEGA 3.5). The relative expression level of salinity tolerance genes (*AeHKT1;4* and *AeNHX1*) were calculated using the 2^−ΔΔCt^ method.

## 5. Conclusions

The present study results determined that Na^+^ movement and exclusion is one of the most imperative physiological attributes of *Ae. tauschii* under salinity stress conditions. *Ae. tauschii* showed salinity tolerance and performed well under salinity stress conditions. Microsatellite markers allied with salinity tolerance showed polymorphism and diversity between these populations. The microsatellite markers *Xgwm 410* and *Xgwm 312* allied with salinity tolerance and showed linkage with salinity tolerance genes (*HKT1;4,*
*HKT1:5*). Our results showed that *HKT1;4* and *NHX1* were synergistically involved in the regulation of Na^+^ by maintaining Na^+^ homeostasis and controlling the Na^+^ movement under salinity stress conditions in *Ae. tauschii*. Under mild salinity stress conditions, *AeNHX1* in leaves compartmentalizes the Na^+^ into vacuoles very slowly, and the Na^+^ concentration starts sequestering in vacuoles, increasing the Na^+^ loading into vacuoles by *AeHKT1;4*. However, under high salinity stress conditions, Na^+^ was increased rapidly and sequestered into vacuoles by *AeNHX1*, and Na^+^ saturated the leaf vacuoles, restricting the Na^+^ transport from roots to leaves and provoking the expression pattern of *AeHKT1;4*. Further, this led to the excessive unloading of Na^+^ from the xylem to alleviate the Na^+^ toxicity from photosynthetic tissues (Figure 7).

In summary, *AeNHX1* and *AeHKT1;4* played an important role in synergistically regulating Na^+^ and controlled the Na^+^ transport in *Ae. tauschii* under salinity stress conditions. Therefore, this synergistic model of salinity is very important and an essential prospect for future application. The salt exclusion mechanism in *Ae. tauschii* has a close genetic relationship with cereal crops (e.g., wheat, barley), so these genes conferring salinity tolerance from *Ae. tauschii* have the potential to be used in the breeding of salt-tolerant cultivars.

## Figures and Tables

**Figure 1 plants-10-01393-f001:**
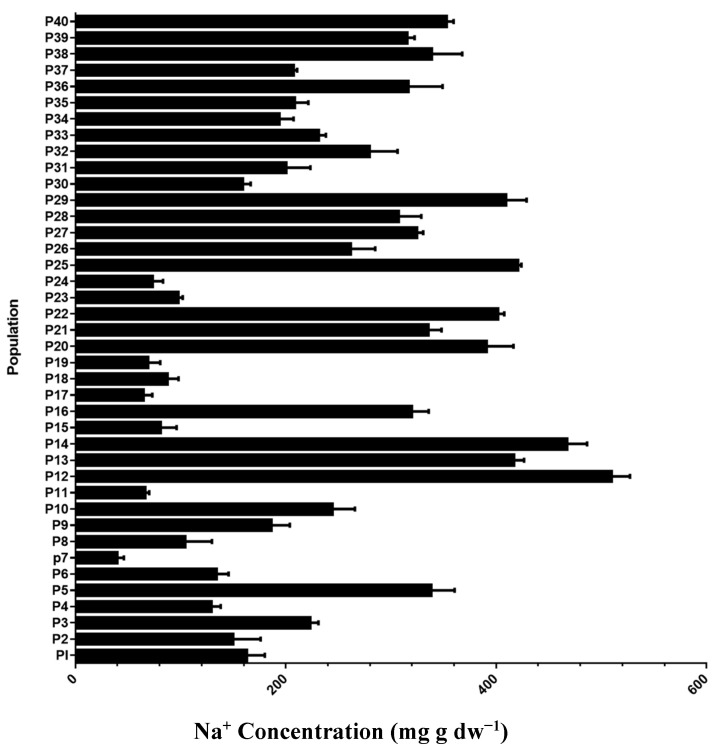
Na^+^ concentration in leaves of 40 *Ae. tauschii* populations under salinity stress conditions (300 mM NaCl). Values are means, and bars indicate standard deviation (SD).

**Figure 2 plants-10-01393-f002:**
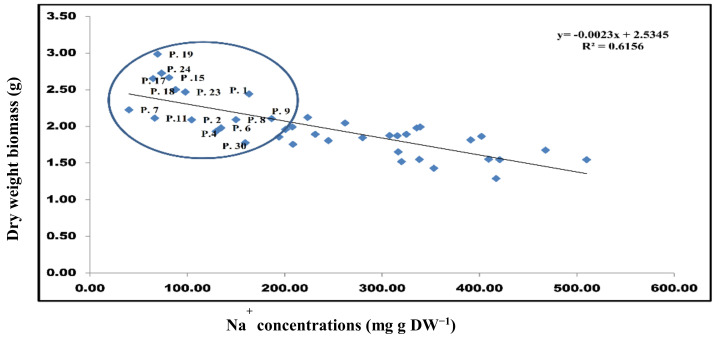
A regression analysis between salinity tolerance and Na^+^ concentrations in 40 *Ae. tauschii* populations were collected from different parts of China. The circle indicates the populations with salinity tolerance, which showed the lowest sodium concentration and maximum dry biomass. The fitted linear regression is R^2^ = 0.62.

**Figure 3 plants-10-01393-f003:**
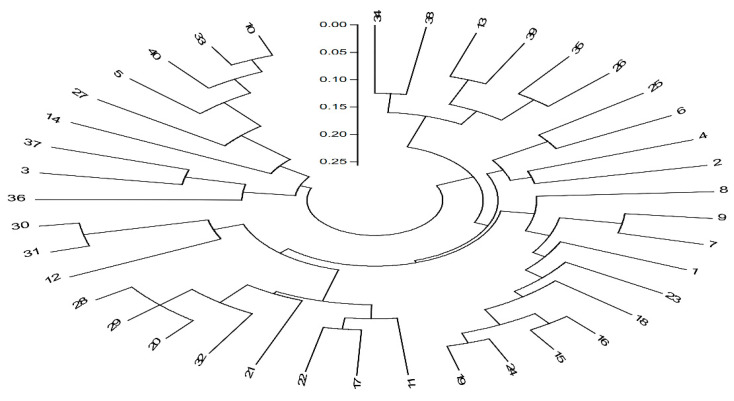
Phylogenetic tree illustrating the genetic relationships among the 40 *Ae. tauschii* populations collected in China based on the 14 SSRs marker.

**Figure 4 plants-10-01393-f004:**
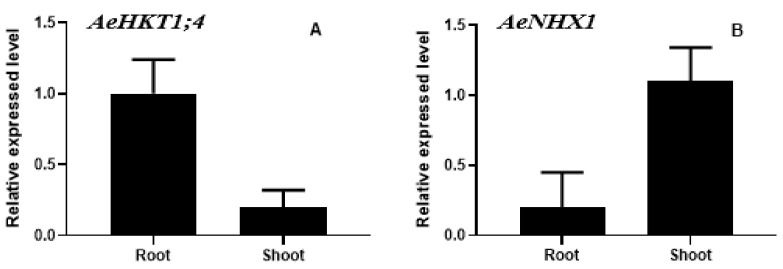
Relative expression levels of *AeHKT1;4* (**A**) and *AeNHX1* (**B**) in roots and shoots of *Ae. tauschii* under control conditions. Actin was used as an internal reference. Experiments were repeated three times. Values are means, and bars indicate standard deviations (SD).

**Figure 5 plants-10-01393-f005:**
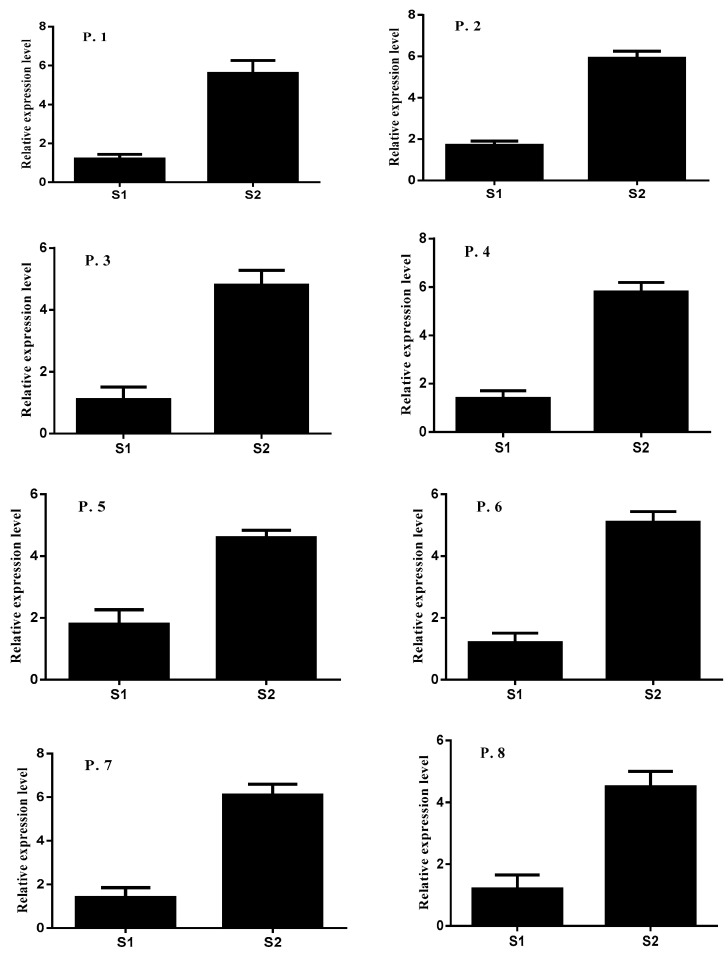
Relative expression levels of *AeHKT1;*4 in in roots of *Ae. tauschii* populations under salinity conditions. Actin was used as an internal reference. Experiments were repeated three times. Values are means, and bars indicate standard deviation (SD). (S1 = 50 mM NaCl and S2 = 200 mM NaCl and P. = population).

**Figure 6 plants-10-01393-f006:**
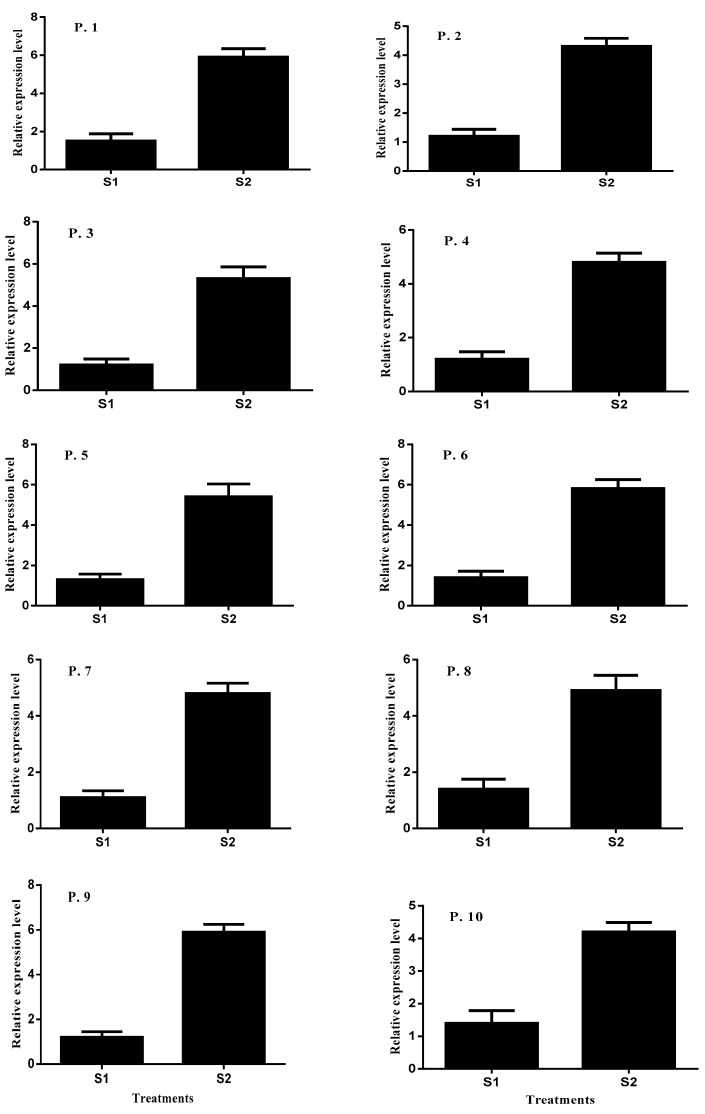
Relative expression levels of *AeNHX1* in leaves of *Ae. tauschii* populations under salinity conditions. Actin was used as an internal reference. Experiments were repeated three times. Values are means, and bars indicate standard deviation (SD). (S1 = 50 mM NaCl and S2 = 200 mM NaCl and P. = population).

**Figure 7 plants-10-01393-f007:**
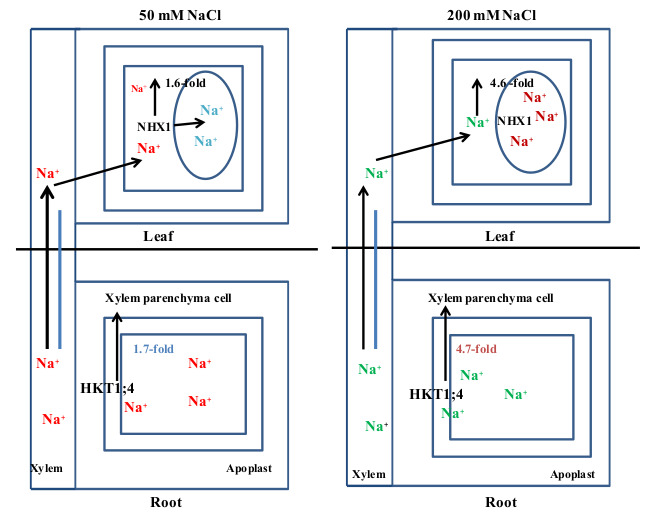
The schematic model of *AeHKT1;4* and *AeNHX1* in Na^+^ transport under salinity stress conditions in *Ae. tauschii*. Under lower salinity stress conditions (50 mM NaCl), *AtNHX1* in leaves compartmentalized of Na^+^ in vacuoles and sequestering Na^+^ would increase loading into xylem by *AeHKT1;4*. Na^+^ could transport in leaves by the transportation stream; under high salinity stress conditions (200 mM NaCl), Na^+^ rapidly and unremittingly sequesters in vacuoles of leaves by *AeNHX1*. The vacuoles’ capacity becomes saturated by sequestering Na^+^, which restricted the Na^+^ transport from roots and induced the expression level of *AeHKT:4* and assisted in the unloading of excessive Na^+^ from the xylem.

**Table 1 plants-10-01393-t001:** Analysis of variance of dry weight biomass (g), plant height (cm), Na^+^, K^+^, and Na^+^/K^+^ ratio under salinity and control conditions.

Source of Variation	df	Dry Weight Biomass	Plant Height	Na^+^	K^+^	K^+^/Na
Replication	2	0.349	2.662	125.63	2793.02	18.38 *
Salt (S)	1	4.516 **	1.443	1035.51 **	9603.59 **	4.61 *
Error 1	2	0.019	9.655	123.56	8936.53	0.23
Population (P)	39	0.444 **	3.614 **	2796.89 **	1535.92 **	18.40 *
S × P	39	0.572 **	4.897 **	2740.92 **	1556.75 **	19.03 **
Error 2	156	0.075	1.873	1032.21	3620.69	10.65
Total	239	0.237	2.720	5911.38	1152.95	13.23

*, ** Significant at 0.05 and 0.01 probability levels, respectively.

**Table 2 plants-10-01393-t002:** Mean values of physiological traits under control and salinity stress conditions.

	Dry Weight Biomass (g)	Plant Height (cm)	Na^+^ (mg g dw^−1^)	K^+^ (mg g dw^−1^)	K^+^/Na
**Control**					
**Mini.**	1.54	10.50	14.67	374.02	14.82
**Maxi.**	3.38	17.00	34.38	693.37	47.91
**Mean**	2.25	13.65	24.73	532.17	23.67
**300 mM NaCl**					
**Mini.**	1.29	10.37	39.97	154.92	0.27
**Maxi.**	2.99	14.89	510.01	680.21	7.56
**Mean**	1.98	13.44	240.71	378.90	1.02

**Table 3 plants-10-01393-t003:** The correlation coefficient between plant dry weight biomass, plant height, Na^+^, K^+^, and K^+^/Na^+^ in *Ae. tauschii* populations (40) under salinity stress condition (200 mM NaCl).

	Dry Weight Biomass	Plant Height	Na^+^	K^+^	K^+^/Na
**Dry weight biomass**	1	0.37 *	−0.784 **	−0.44 **	0.41 **
**Plant height**		1	−0.3127 *	−0.063	0.17
**Na^+^**			1	0.411 **	−0.56 **
**K^+^**				1	0.36 **
**K^+^/Na^+^**					1

*, ** Significant at 0.05 and 0.01 probability levels, respectively.

**Table 4 plants-10-01393-t004:** Major allele frequency, allele number, gene diversity, and polymorphism information content of the 40 *Ae. tauschii* populations collected from China based on the 14 SSRs.

Marker	Major Allele Frequency	Allele No	PIC
Xgwm428	0.73	3	0.34
Xbarc159	0.89	2	0.18
Xgwm205	0.95	3	0.09
Xgwm55	0.95	3	0.09
Xgwm312	0.36	8	0.72
Xgwm3	0.36	8	0.72
Xbarc273	0.41	7	0.73
Xgwm410	0.15	8	0.92
Xgwm165	0.73	3	0.34
Xwmc773	0.73	3	0.39
Xbarc74	0.73	3	0.39
Xgwm609	0.98	2	0.05
Xgwm583	0.95	3	0.09
Xwmc367	0.58	4	0.45
Mean	0.68	4.28	0.39

**Table 5 plants-10-01393-t005:** Sequence of 14 simple sequence repeats (SSR) primers.

Sr.No	Primers	Forward Primer (5′ to 3′)	Reverse Primer (5′ to 3′)	5′ Modify
1	Xgwm 583	TTCACACCCAACCAATAGCA	TCTAGGCAGACACATGCCTG	Hex
2	Xwmc 773	GAGGCTTGCATGTGCTTGA	GCCAACTGCAACCGGTACTCT	6-Fam
3	Xbarc 74	GCGCTTGCCCCTTCAGGCGAG	CGCGGGAGAACCACCAGTGACAGAGC	Hex
4	Xbarc 159	CGCAATTTATTATCGGTTTTAGGAA	CGCCCGATAGTTTTTCTAATTTCTGA	6-Fam
5	Xbarc 273	AATTCAGAGAAACACACCTCCCTTTTA	ACTCCATCAACCCCGTTCATT	Hex
6	Xwmc 367	CTGACGTTGATGGGCCACTATT	GTGGTGGAAGAGGAAGGAGAGG	6-Fam
7	Xgwm 428	CGAGGCAGCGAGGATTT	TTCTCCACTAGCCCCGC	Hex
8	Xgwm 609	GCGACATGACCATTTTGTTG	GATATTAAATCTCTCTATGTGTG	6-Fam
9	Xgwm 55	GCATCTGGTACACTAGCTGCC	TCATGGATGCATCACATCCT	Hex
10	Xgwm 312	AGGAGCTCCTCTGTGCCAC	TTCGGGACTCTCTTCCCTG	6-Fam
11	Xgwm 3	GCAGCGGCACTGGTACATTT	AATATCGCATCACTATCCCA	Hex
12	Xgwm 410	GCTTGAGACCGGCACAGT	CGAGACCTTGAGGGTCTAGA	6-Fam
13	Xgwm 205	CGACCCGGTTCACTTCAG	AGTCGCCGTTGTATAGTGCC	6-Fam
14	Xgwm 165	TGCAGTGGTCAGATGTTTCC	CTTTTCTTTCAGATTGCGCC	Hex

**Table 6 plants-10-01393-t006:** Sequence of primers used for real-time PCR amplification.

Primer	Sequences (5′–3′)	Gene
P1	GCGTTCTTGTGCTTCTTG	*AeACTIN-F*
P2	TTCTGACCTTGACCATTCC	*AeACTIN-R*
P3	ACGCGCTCAAAATGTAACCG	*AeHKT1;4 F*
P4	TGCCAAATCAAGGGCTCCAA	*AeHKT1;4-R*
P5	CGGCAGTGCATGAAACTGTG	*AeNHX1-F*
P6	TTTTCTCCGGTTATGCCGCT	*AeNHX1-R*

## Data Availability

The data presented in this study are available in Appendix A.

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
