# Peer review of "Genetic Diversity and Synergistic Modulation of Salinity Tolerance Genes in Aegilops tauschii Coss"

_plants, 2021, doi:10.3390/plants10071393_

Round 1

Reviewer 1 Report

The authors answered the requests, however, the manuscript still needs the revision of the English form

Reviewer 2 Report

In Table 2, Figures 1 and 2,  

The Na+ and K+ concentration is not realistic. It is too much high.

For example, in population 12, Na+ concentration is more than half of the dry weight.

In figure 3,

"*, ** Significant at 0.01 and 0.05" 

should be changed to 

"*, ** Significant at 0.05 and 0.01"

Reviewer 3 Report

Although some points and suggestions (included in  previous reviewer comments)  have been revised, the manuscript  “Genetic diversity and synergistic modulation of salinity tolerance genes in Aegilops tauschii Coss.“ still shows some typos and grammar errors, and  mainly  there are still some fundamental concerns, with the statistical analysis procedure (already highlighted in previous reports) and an unclear and  incomplete  representation of the data as:  the analysis of variance of measured traits among the Aegilops population in the control and salinity treatments; “significant value” of HKT1;4 and NHX1 expression levels.  The paragraph “4.4 Data analysis” should be include a description of statistical analysis procedures for physiological Traits, regression and gene expression analysis.
I therefore suggest reconsidering the drafting of the manuscript by completing the presentation of data and and reporting the statistical procedures in exhaustive way to give proper weight to your conclusions

Round 2

Reviewer 3 Report

The  last version of manuscript “Genetic diversity and synergistic modulation of salinity tolerance genes in Aegilops tauschii Coss.” reports still unclear results and makes difficult to follow the presented data. A deeper  analysis  on each population is suggested 

Author Response

Dear reviewer.

We thank you for your useful comments and suggestions, which have significantly improved the manuscript. Your points and suggestions help us to improve our manuscript. We carefully improve the English of our manuscript. We hope this revised manuscript will be acceptable for publication in plants.

question by reviewer; The  last version of manuscript “Genetic diversity and synergistic modulation of salinity tolerance genes in Aegilops tauschii Coss.” reports still unclear results and makes difficult to follow the presented data. A deeper  analysis  on each population is suggested 

Response:

We add complete details regarding Genetic diversity and synergistic modulation of salinity tolerance genes in Aegilops tauschii Coss.

We add complete details of physiological parameters through ANOVA, regression and Correlation analysis in Table 1,2,3  and figure 1,2

We add details regarding genetic diversity in table 4,5 and figure 3

We use 11 population for gene expression and their details in figure 4.5 and 6

We make a synergistic model of the salinity tolerance gene in figure 7.

A similar study conducted in many other species and published in noted journals; here, we attached one manuscript.  Title (Salinity tolerance of Aegilops cylindrica genotypes collected from hyper-saline shores of Uremia Salt Lake using physiological traits and SSR markers) (DOI: 10.1007/s11738-014-1602-0). Even this kind of studies do not add information regarding population gene expression data, but we add data regarding population gene expression data in the manuscript on reviewers demands.

This is all about information of this manuscript which we add in manuscript, and we do not conduct further analysis on these population.

We hope this revised manuscript will be acceptable for publication in plants. Thanks.

Regards

Xiangju Li, Professor, Ph.D.

Weed Science
Institute of Plant Protection (IPP)
Chinese Academy of Agricultural Sciences (CAAS)
No.2 West Yuanmingyuan Road
Haidian
Beijing 100193
China
Tel./Fax. 86 10 62813309; Tel. 86 10 62813309
[email protected]

Round 3

Reviewer 3 Report

The authors of manuscript “Genetic diversity and synergistic modulation of salinity tolerance genes in Aegilops tauschii Coss.“ have considered and answered the questions reported in previous review.

This manuscript is a resubmission of an earlier submission. The following is a list of the peer review reports and author responses from that submission.

Round 1

Reviewer 1 Report

The paper “Genetic diversity and synergistic modulation of salinity tolerance genes in Aegilops tauschii Coss.” reports an analysis at physiological and molecular level of several  Aegilops genotypes showing contrasting salt tolerance. The aim was to understand the role of two genes AeHKT14 and AeNHXI in salt stress response. Then there is also an analysis with microsatellites on the set of genotypes utilised with aim to find markers associated with salinity tolerance

The topic is interesting, however, I found several drawbacks and I think the manuscript in the present form is not suitable for publication.

The phenotypic characterization of the set of Aegilops genotypes is poor and based only on the evaluation of the dry biomass and Na+ content.

What you call population are different cultivars or landraces? Are they sharing common ancestors? Please add more details.

The choice of the marker set was based on their genomic location? Are they representative of the whole genome?

The number of Populations  utilized is limited so it is difficult to find a robust association between the tolerant phenotype and a marker or set of markers, since also the set of markers is very limited.

I don’t see a clear demonstration of the association between markers and tolerant phenotype as claimed in the conclusion (The microsatellite markers Xgwm410 and Xgwm314 allied with salinity tolerance and showed linkage with salinity tolerance genes (HKT1;4, HKT1:5).)

Similarly the comment reported about expression analysis are not precise (a 1.7 fold change cannot be considered relevant) and furthermore statistical analysis is completely missing.

The English form needs to be improved.

In fig. 2 the x- and y- axes to not have a label

In fig. 5 what is S1 and S2, no explanation is given in the caption to fig. 5

Reviewer 2 Report

The authors tried to evaluate the genetic diversity and salinity tolerance using physiological traits and microsatellite markers in Aegilops tauschii. They measured the expression of the genes related to salinity tolerance using real-time quantitative PCR (qRT-PCR).

Nevertheless, the following issues should be revised to be considered for further review.

In Table 3 and 4, Unit of NaCl concentration should be corrected.

In table 4, the meaning of * and ** should be described.

In figure 2, there is no legend for X and Y axis.

In figure 4, they were confused shoot and leaves.

In figure 5, I cannot understand what they are describing.

Reviewer 3 Report

The manuscript “Genetic diversity and synergistic modulation of salinity tolerance genes in Aegilops tauschii Coss.“reports physiological and molecular analysis in order to evaluate the genetic diversity and salinity tolerance in Aegilops tauschii. Although the subject covered by the manuscript is important to achieve sustainable agricultural approaches, in view of increasing salinization of arable land, the work presents several main problems.

Below are some points of the manuscript to be reviewed, but overall the work needs a thorough rewriting after further increase of the data collected and their statistical anlysis

The abstract should be better structured in order to give a condensed information of results reported in the full text of research manuscript and needs correction in some terms.

Here some example

Line 12 : …. tolerance in Aegilops tauschii. The scientific names of species in Italics (check all text)

Line 12: . The molecular  marker…… before starting with the results a brief description of the materials and methods used would be desirable

Line 15: ….  and AeNHXI…is AeNHX1 (check in all manuscript)

The introduction fails to frame clearly and comprehensively the items of the work. In several instances I also suggest to cite more relevant and recent literature. The introduction, therefore, should be revised in the flow of concepts to better explain the aim of the study.

Here some examples:

Line 29 and line 31: update the references

Line 34: it not clear is the citation [3] in this sentence

Line 34-35 : Salt stress inhibits plant development, restricting photosynthesis.

This is redundant  

Line 37:…..the cell aided by NHX1……

Abbreviations should be defined in parentheses the first time they appear in the abstract (all manuscript)

Line 40-54: I suggest to complete the background description of impact soil salinity in plants (research area) and then to move on the Ae. Tauschii information and relationship with wheat.

Line 44-50: sentences should be reworked in order to improve it, and to correct the references and bring it up to date (missing SSR AFLP)

Line 70-line 78 : check the right position of citations

Line 82-83: ….. we evaluated the salinity tolerance populations …..

the authors should more clearly state the materials used  

Line 84-85: In the second phase, we analyzed the expression profile of AeHKT1;4 and AeNHX1 in Ae. tauschii under different salinity treatments.

as indicated above

Materials and methods

Physiological characterization needs to be improved and data on plant growth traits would be necessary to add.

Four physiological traits are collected only from leaves.  In my opinion, Na+ and K+ concentration would also have been desirable measurement in roots.

Line 457: Out of 26 primers pairs, ……..

Which primers? More information  needed about the choice of those specific SSRs                                    

Line 498: Total RNA was extracted by using…….

which samples were used for the RNA extraction?

The statistical analysis is completely missing and this invalidates any evaluation and discussion of the reported results.

Results

Line 90-103: These data are not showed in table.

(no statistical analysis as discussed above)

Line 95: Significant variations among populations were observed in Na+ and K+ accumulation……

Analysis of variance (ANOVA) was not done and then it is no possible to report significant differences.

Any tables for analysis of variance (mean squares) of data is not reported

Line 136-139: On the base of cluster analysis….

statistical analysis is missing.

Figure 3: How was  phylogenetic analyses conduct?

Line246-246:  It is no specified which population  was used for RNA extraction and then for qRT-PCR analysis

Discussion and Conclusion

 In light of the above considerations (especially for statistical analysis), discussion a conclusion have to rewrite

Round 2

Reviewer 1 Report

The manuscript still presents various problems, the principal concerns the English form which is not acceptable for publication in a scientific journal. Authors should have a native speaker reviewer. There are also many typos.

The caption of Figure 5 is incorrect. It reports results of expression analysis in salt stress conditions, not in control conditions.

In line 511 it is written ‘We used 11 populations that perform better under salinity stress conditions’ but I do not see the results of expression analysis reported for eleven populations, only for one.

Then it is reported (line 518-519) that ‘The relative expression level of salinity tolerance genes was calculated by using the 2−ΔΔCt method.’

You should specify in any analysis which was the sample used as calibrator to normalise the data.

Still, there is not a paragraph (in Materials and method section) about  the statistical analysis performed for the different data sets.

Reviewer 2 Report

In table 4, please add a foot note to describe the statistical significance of the data displayed in the symbols * and **.

In figure 2, there is no legend for X and Y axis in the figure itself.

Round 3

Reviewer 1 Report

I do not see any improvement, there are still many typos and grammar error. The experiment of gene expression is not adequately reported and I suggest completing this analysis on each population without averaging all the data, this could be more informative.

Author Response

Reviewers report 1.

Point 1; I do not see any improvement, there are still many typos and grammar error. The experiment of gene expression is not adequately reported and I suggest completing this analysis on each population without averaging all the data, this could be more informative.

Response

We thank you for your useful comments and suggestions, which have significantly improved the manuscript. We have revised the manuscript and incorporated all the suggestions. Your points are valid, but our gene expression studies under process, and your suggestions help us incoming articles related to this study. Thanks.

We believe data in this manuscript is enough for readers to understand our idea of this publication. We have further experimented with these populations and use these populations in transcriptome and metabolomics analysis and after this, we have a plan to check different gene expression analyses. We believe this publication has enough data to understand our idea. We humble request to editor and reviewer to ignore this point. We will be very thankful for this.

Regards

Xiangju Li, Professor, Ph.D.

Weed Science
Institute of Plant Protection (IPP)
Chinese Academy of Agricultural Sciences (CAAS)
No.2 West Yuanmingyuan Road
Haidian
Beijing 100193
China
Tel./Fax. 86 10 62813309; Tel. 86 10 62813309
[email protected]

Reviewer 3 Report

Comments and Suggestions for Authors

The authors of “Genetic diversity and synergistic modulation of salinity tolerance genes in Aegilops tauschii Coss.“ paid heed to most of advice and suggestions and the manuscript was revised. Moreover, the manuscript should be undergone to minor revision. Below is a summary of the recommended changes (see the green highlight in the attached manuscript):

Point 1:

correct AeNHXI in AeNHX1 (check  manuscript)

Point 2:

Line 46: . To determine genetic diversity,…. this part can be removed because it is reported at the end of sentence

Point 3:

Line 77-78: This description was reported in the sentences from line 40 to line 51; then I suggest to  move Line 40-line 51 here. In this way  the flow of information sounds better and it is not necessary to repeat the information already given 

Point 4:

line 127: ** showed significance level.

specify
 * = Correlation  is  significant  at  
** =Correlation  is  significant  at  

Point 5:

The tables of manuscript must be numbered following their number of appearance as reported in the “Instructions for Authors”. Check the order of appearance of the tables and figures within the manuscript and renumber them. Also try to place them as close as possible to their first citation.

Point 6:

Figure 2: the unit of measure ,Na+ concentrations and Salinty level, should be inserted

Point 7:

Figure 5: I think that ….(no additional NaCl) ….. should be deleted. Probably it’s a typo.

Point 8:

Line 340: Sixty alleles were amplified in sixteen primers, with an average….

Table 5 reports 14 primers no sixteen, as also described in Materials and methods

Point 9:

Line 385: “….. by increasing Na+ concentration [44] (Bao et al., 2014).”

Delete “(Bao et al., 2014)

Point 10:

line 553-554 “The microsatellite markers Xgwm410 and Xgwm314 allied 553 with salinity tolerance and showed linkage with salinity tolerance genes (HKT1;4, 554 HKT1:5)”

 in line 351 (Discussion) the authors report "The microsatellite markers Xgwm 312 and Xgwm 410 are linked with salinity tolerance and associated with salinity the tolerance gene HKT1;5 and HKT1;4 in wheat crops "

check the right primers: “Xgwm 410 and Xgwm 314” or “Xgwm 312 and Xgwm 410” ?

Point 11:

Line 547: Table 2  P=primer it is not necessary. The header contains column label.

Point 12:

Pay attention to the references. See  the green highlight
